# Contact Allergy to Castor Oil, but Not to Castor Wax

**Michel Verheyden, Sven Rombouts, Julien Lambert and Olivier Aerts \***

Department of Dermatology, University Hospital Antwerp (UZA) and University of Antwerp (UAntwerpen), B-2650 Antwerp, Belgium; michel.verheyden89@gmail.com (M.V.); svenrombouts@hotmail.com (S.R.); julien.lambert@uza.be (J.L.)

**\*** Correspondence: olivier.aerts@uza.be; Tel.: +32-3-821-4272; Fax: +32-3-825-3428

**Abstract:** *Ricinus communis* (castor) seed oil (CAS 8001-79-4), a vegetable oil extracted from the seeds of *Ricinus communis*, is widely used in cosmetics and pharmaceuticals, and may be a cause of allergic contact dermatitis from these products. We present two patients with allergic contact dermatitis from cosmetics containing castor oil, in whom a correct diagnosis was achieved by patch testing castor oil 'as is'. PEGylated and/or hydrogenated derivatives (the latter formerly also available from patch test allergen suppliers) and/or cosmetics containing these specific derivatives did not result in contact allergy or allergic contact dermatitis. This observation might be relevant for the manufacturing of cosmetics and pharmaceuticals. In the future, further research into the allergenicity of castor oil and its numerous derivatives, and their optimal patch test concentrations, may be desirable.

**Keywords:** allergic contact dermatitis; CAS 8001-79-4; castor oil; castor wax; cera alba; cheilitis; cosmetics; hydrogenation; hydrogenated castor oil; lipbalm; PEGylation; ricinus oil; *Ricinus communis* (castor) seed oil

## 1. Introduction

*Ricinus communis* (castor) seed oil (CAS 8001-79-4; "castor oil") is a yellowish, vegetable oil extracted from the seeds of *Ricinus communis* (*Euphorbiaceae*). It is a triglyceride and 90% of its fatty acid content consists of ricinoleic acid; oleic acid and linoleic acid are other important components. Hydrogenation, that is, the addition of hydrogen to castor oil in the presence of a nickel catalyst, results in hydrogenated castor oil (*syn.* castor wax) which is hard and brittle, but retains its lubricity, polarity and surface wetting properties, valuable for many industries [1–3]. The component 12-hydroxystearic acid is the principal fatty acid in this particular castor oil-derivative.

Ricinoleic acid and 12-hydroxystearic acid are also believed to be the potential allergens of castor oil and hydrogenated castor oil, respectively [4].

Cases of allergic contact dermatitis from the oil, and from its derivatives, have occasionally been reported [5–7]. We here report 2 additional cases of allergic contact dermatitis from castor oil present in lip balms, whereas patch tests with hydrogenated castor oil 30% in petrolatum (pet.) did not elicit skin reactions.

## 2. Case Reports

### 2.1. Patient 1

A 25-year-old atopic woman suffered from perioral eczema which she related to the use of several lip balms. Moreover, a facial rash had occurred in the past, following the use of a day cream. Patch tests were performed on the upper back with the Belgian baseline series and a cosmetic series (both from Chemotechnique®, Vellinge, Sweden). Castor oil, obtained from the hospital pharmacy, is included in our cosmetic series and was patch tested 'as is' (100%), whereas hydrogenated castor oil (obtained from

the same cosmetic manufacturer as in Patient 2, see below), was tested at 30% pet. Also, the patients' own cosmetic products, and their available ingredients, were patch tested. Patch test chambers were from Allergeaze® (SmartPractice, Calgary, AB, Canada). Following an occlusion of 2 days, readings were performed according to the guidelines of the European Society of Contact Dermatitis (ESCD) [8]. At day (D) 4 positive reactions were seen to castor oil ('as is') (+), and to cera alba 30% pet. (++), both present in the lip balms she had been using. Furthermore, she reacted to dexpanthenol 5% pet., also present in two lipbalms, and in the moisturizing cream which had provoked a facial dermatitis in the past. Other positive reactions included fragrance mix II, sodium metabisulphite, decyl glucoside and nickel. There was no patch test reaction to hydrogenated castor oil 30% pet; higher test concentrations for hydrogenated castor oil had not been applied. Ricinoleic acid and 12-hydroxystearic acid could not be patched tested separately. The perioral dermatitis completely healed upon avoidance of all contact allergens.

### 2.2. Patient 2

A 50-year-old male patient presented with a perioral eczema following the use of two different brands of lip balms. Initial patch tests (performed as in Patient 1) were positive to castor oil ('as is') (++). A patch test with the lip balms could not be performed at that time, but the patient performed a repeated open application test (ROAT) with one of two lipbalms, twice daily on a forearm, which resulted in a positive reaction after three days. Upon request, the manufacturer of this particular lip balm sent us the individual ingredients, among which were castor oil ('as is') and hydrogenated castor oil (30% pet.). Positive reactions were again seen to castor oil 'as is' (++ at D2 and +++ at D4, respectively) (Figure 1), but not to hydrogenated castor oil 30% pet; higher test concentrations for hydrogenated castor oil were not used. Ricinoleic acid and 12-hydroxystearic acid could not be patched tested separately. Castor oil was, according to the packaging, present in both lip balms he had been using. The patient also showed positive patch test reactions to benzyl salicylate, explaining a previous axillary dermatitis from deodorant, and to epoxy resin, for which no relevance was found. Interestingly, the patient noticed he was able to tolerate an after-shave cream containing a polyethylene glycol (PEG) computation of castor oil, that is, PEG-40 hydrogenated castor oil. Following avoidance of all contact allergens, and of castor oil in particular, the patient's perioral eczema healed and it has not recurred since.

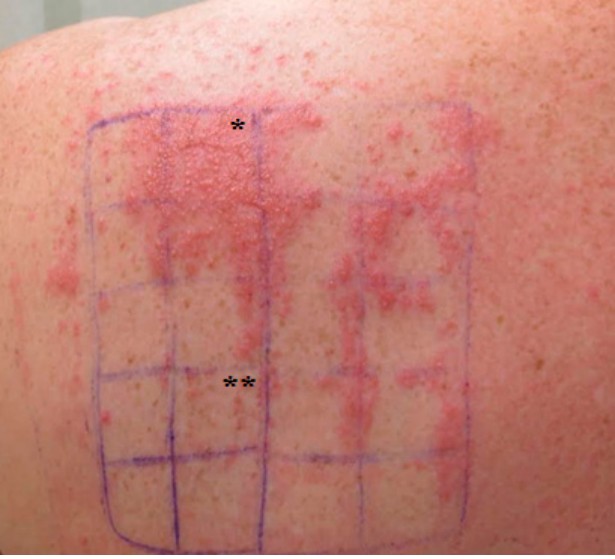

**Figure 1.** Strong positive patch test reaction (+++) to castor oil ('as is') (*), but not to castor wax (30% pet.) (**) on D4 in Patient 2. Note the strong spreading reaction.

## 3. Discussion

Castor oil, because of its high viscosity, its solubility for most pigments, and its masking, perfuming and skin conditioning properties, is often used in cosmetics, such as lipsticks, sunscreens, deodorants, moisturizers, nail lacquer removers and soaps [9]. It is also used in pharmaceuticals (e.g., in laxatives) [5], as biofuel [10], and as a lubricant and for food flavoring [1,3,5].

Many derivatives exist, such as acetylated, hydrogenated and PEGylated forms, even in combinations, although little is known about the allergenic potential of these variants in patients contact-allergic to castor oil. Castor oil, hydrogenated castor oil, and other derivatives, have all been found to be a possible cause of allergic contact dermatitis [4,11,12]. Also, the stems, leaves and seeds of the Ricinus plant may cause occupational allergic contact dermatitis, particularly in farmers harvesting the plants, or in laboratory workers exposed to them [13]. In our department, castor oil is routinely patch tested 'as is', as part of a cosmetic series, but the vast majority of patients (~400 every year, also serving as control patients for the two cases reported here) show no allergic or irritant reactions to this allergen.

Kalavala et al. previously reported PEG-7 hydrogenated castor oil as a cause of allergic contact dermatitis, and their report suggested that higher molecular weight PEGylated castor oil-derivatives may have less potency to cause allergic contact dermatitis [12]. In both our patients hydrogenated castor oil (30% pet.) did not lead to a positive patch test reaction, and in the second case a PEGylated (PEG-40) and hydrogenated derivative, as an ingredient of an after-shave cream, did not provoke any skin reaction.

One study, as described in [14,15], concerning twenty patients, found no contact allergy to hydrogenated castor oil 30% pet. However, Taghipour et al. reported a patient who showed a false-negative patch test reaction to hydrogenated castor oil 30% pet. whereas hydrogenated castor oil 'as is' (100%) did result in a positive reaction; castor oil was not separately patch tested 'as is' [11]. We were not able to patch test hydrogenated castor oil 'as is' (100%) in our patients, and this might still have produced a positive patch test according to [11], although in our second patient a PEGylated and hydrogenated castor oil derivative, present in another cosmetic, appeared to be well tolerated. Hydrogenated and/or PEGylated derivatives, as they are less reported allergens [4], might perhaps be more weak allergens, although this remains to be proven. Although ricinoleic acid and 12-hydroxystearic acid are considered to be the principal allergens in castor oil and in hydrogenated castor oil, respectively, also impurities might theoretically still be the actual allergens.

Our cases illustrate that, in order to achieve a correct diagnosis, one should patch test castor oil 'as is', and not rely on patch test materials containing only hydrogenated derivatives, as were formerly obtainable from allergen suppliers (e.g., hydrogenated castor oil 20% pet., from Chemotechnique®), or as supplied by cosmetic manufacturers (e.g., hydrogenated castor oil 30% pet.).

## 4. Conclusions

Castor oil may be responsible for allergic contact dermatitis from cosmetics, and can be patch tested separately 'as is'. PEGylated and/or hydrogenated derivatives might sometimes be tolerated in these patients, an observation which may be relevant for the cosmetic and pharmaceutical industry. In the future, further research into the allergenicity of castor oil and its numerous derivatives, and their optimal patch test concentrations, may be desirable.

**Author Contributions:** Each author has participated sufficiently to take public responsibility for appropriate portions of the work and consented to the final, submitted version.

**Conflicts of Interest:** The authors declare no conflict of interest.

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
