# Peer review of "Contact Allergy to Castor Oil, but Not to Castor Wax"

_cosmetics, doi:10.3390/cosmetics4010005_

Round 1

Reviewer 1 Report

An interesting report of two cases highlighting the importance of patch testing with the correct castor oil derivative.

Broad comments

Some information on the composition of castor oil/ hydrogenated castor oil could be added. What are the main constituents? How does the composition of the hydrogenated castor oil differ from that of the ‘untreated’ castor oil? What allergens have been described in castor oil and hydrogenated castor oil? Some substances are mentioned on line 73-74, but it is not explained whether these are ingredients of castor oil and/or hydrogenated castor oil.

The report can be improved with a discussion on why the hydrogenated castor oil did not give positive patch test reactions. Could it be explained by the hydrogenation of fatty acids, by differences in content of other substances or impurities or by the different patch test concentrations used?

It is mentioned in the text that castor oil was tested as a part of the cosmetics series and also that both patients’ products contained castor oil. However, it is not mentioned why the authors chose to test also with hydrogenated castor oil.

Preferably, the INCI names should be used for all cosmetic ingredients including the castor oil derivatives.

The format of the reference list needs to be improverd. 4 references in the list are not cited in the text.

Specific comments

Line 38: What patch test chambers were used?

Line 48 (and fig 1): Fig 1 shows a strong reaction spreading outside the test area. Could the reactions to substances tested next to castor oil be evaluated? Were they re-tested? The exact localization of castor oil could be indicated in the figure.

Line 50: Was the ROAT performed with both lip balms?

Line 62: It is not obvious what ‘pegylated’ refers to. Should it be written ‘PEGylated’?

Line 74: Is the correct study cited?

Line 82: What does ‘c.q’ mean?

References: Check the format of all references

References 15, 16, 17 and 18 are not cited in the text. Should they be removed??

Author Response

Dear Editor, dear Reviewer #1,

We thank you for your questions and comments; these allow us to improve the manuscript.

To answer your questions:

1.Some information on the composition of castor oil/ hydrogenated castor oil could be added. What are the main constituents? How does the composition of the hydrogenated castor oil differ from that of the ‘untreated’ castor oil?

Castor oil is a triglyceride and 90% of its fatty acid content is composed of the unsaturated ricinoleic acid; oleic acid and linoleic acid are other important components. Hydrogenation of castor oil, that is, the addition of hydrogen to castor oil in the presence of a nickel catalyst, results in hydrogenated (saturated) castor oil, as such being one of many derivatives of castor oil. The component 12-hydroxystearic acid is the principal fatty acid in hydrogenated castor oil.

This information was added to the text.

2.What allergens have been described in castor oil and hydrogenated castor oil?

The actual allergens of castor oil and hydrogenated castor oil remain elusive, but it is believed that ricinoleic acid and 12-hydroxystearic acid  are the likely allergens.

This information was added to the text.

3.Some substances are mentioned on line 73-74, but it is not explained whether these are ingredients of castor oil and/or hydrogenated castor oil.

This line, which is indeed confusing, will be rephrased to:

One study, as described in [14] and [15], concerning twenty patients, found no contact allergy to hydrogenated castor oil 30% pet.

4.The report can be improved with a discussion on why the hydrogenated castor oil did not give positive patch test reactions. Could it be explained by the hydrogenation of fatty acids, by differences in content of other substances or impurities or by the different patch test concentrations used?

Hydrogenation might perhaps attenuate (weaken) the allergenic potential of castor oil, although this remains to be proven. Differences in reactivity to castor oil and to hydrogenated castor oil, respectively, might also be explained, for example, by impurities (acting as allergens) present in castor oil and in its hydrogenated counterpart. Finally, it might also be that the hydrogenated castor oil 30% pet. patch test was false negative, as it has been reported that hydrogenated castor oil 100% can be positive while 30% remains negative; however, this does not fully explain why our first patient did not react to cosmetics containing pegylated and hydrogenated castor oil.

This information was added to the text.

5.It is mentioned in the text that castor oil was tested as a part of the cosmetics series and also that both patients’ products contained castor oil. However, it is not mentioned why the authors chose to test also with hydrogenated castor oil.

In Patient 2 castor oil and hydrogenated castor oil were both present in one lipbalm that led to allergic contact dermatitis. When patch testing the individual components, a reaction to castor oil (‘as is’; 100%) was seen, but not to hydrogenated castor oil 30% pet. When Patient 1 was seen (in time after Patient 2), and was shown to be reacting to castor oil, we decided to patch test hydrogenated castor oil 30% pet. in her as well.

6.Preferably, the INCI names should be used for all cosmetic ingredients including the castor oil derivatives.

The correct INCI names are:

- Ricinus communis (castor) seed oil (instead of ‘castor oil’)

- Hydrogenated castor oil (instead of ‘castor wax’)

For linguistic and practical reasons, we would suggest keeping the names ‘castor oil’ and ‘hydrogenated castor oil, and ‘castor wax’ in the title, but mentioning the correct INCI nomenclature in the introduction of the manuscript.

7.The format of the reference list needs to be improved, 4 references in the list are not cited in the text.

The format was adjusted and the non-cited (and non-relevant) references, which were accidentally included, were removed.  

Specific comments

1.Line 38: What patch test chambers were used?

Patch test chambers were from Allergeaze® (SmartPractice, Calgary, Canada). This was added to the text.

2.Line 48 (and fig 1): Fig 1 shows a strong reaction spreading outside the test area. Could the reactions to substances tested next to castor oil be evaluated? Were they re-tested? The exact localization of castor oil could be indicated in the figure.

The exact localization of castor oil and of hydrogenated castor oil are now indicated in the Figure, with an asterix (*) and double asterix (**), respectively.

Only castor oil gave a ++ reaction at D2, which escalated to a +++ reaction at D4; the strong spreading reaction was considered by us to be the result of the strong reactivity to castor oil; the patch test underneath is liquid paraffin (this was not re-tested). Of note, this already concerned a re-testing of the patient with the individual ingredients of one of the lipbalms, as obtained from the cosmetic manufacturer; a ‘booster’ effect, resulting in stronger reactivity during the second patch test series as compared to the first initial patch test series, may perhaps not be excluded.

This was detailed in the text.

3.Line 50: Was the ROAT performed with both lip balms?

Only with one of two lip balms (the one containing both castor oil and hydrogenated castor oil) a ROAT was performed (as indicated in the text), and this gave a positive result.

4.Line 62: It is not obvious what ‘pegylated’ refers to. Should it be written ‘PEGylated’?

Pegylated refers to a polyethyleneglycol computation of a substance, in this case of castor oil. To avoid confusion we agree to write PEGylated throughout the text, instead of pegylated.

5.Line 74: Is the correct study cited?

The study cited is a review (dating from 2007) on published studies on castor oil and derivatives; the data mentioned on hydrogenated castor oil refer to the patch test book by De Groot et al (De Groot, A. C., ed. 1994. Patch testing. Test concentrations and vehicles for 3700 Chemicals, 2nd ed., 11, 145. Amsterdam: Elsevier.). We cited both.

6.Line 82: What does ‘c.q’ mean?

Casu quo, which may refer to ‘and’ or to ‘or’, but, to avoid confusion we rephrased to: an observation which may also be relevant for the cosmetic and pharmaceutical industry.

7.References: Check the format of all references

The format was checked, and adjusted accordingly.

8.References 15, 16, 17 and 18 are not cited in the text. Should they be removed??

These references were accidentally withheld in the reference list, but were not considered necessary; they were removed.

Reviewer 2 Report

Verheyden et al report two cases of castor oil. Given the wide use of castor oil in cosmetics it is of interest to the community to know about the potential issue of castor oil contact dermatitis. 

Issues identified

1) It is not clear if negative control patients were used? How often is castor oil used in patch testing and how often are positives seen?

2) The Figure presented needs to have a legend to explain which squares are which allergen. Also it appears that many of the areas are reactive

3) The castor oil reaction might be irritant in nature and not allergic. A dose response in petrolatum might help to resolve this or to at least test castor oil at the percentage found in the cosmetics in question. 

Author Response

Dear Editor, dear Reviewer #2,

We thank you for your questions and comments; these allow us to improve the manuscript.

To answer your questions:

1) It is not clear if negative control patients were used? How often is castor oil used in patch testing and how often are positives seen?

Castor oil is part of our cosmetic series and is patch tested ‘as is’ in patients experiencing skin problems from cosmetics/topicals (i.e. it is patch tested ‘as is’ in ~ 400 patients every year). In the vast majority (and in any case in > 20 controls) no allergic or irritant reactions are seen. This was added to the text.

2) The Figure presented needs to have a legend to explain which squares are which allergen. Also it appears that many of the areas are reactive.

Castor oil and hydrogenated castor oil are now detailed on the Figure. Many areas are indeed reactive on D4; on D2 only castor oil itself reacted (++) and this escalated towards D4 (+++). This is possibly explained by the fact that it concerned a re-testing of this patient with all ingredients of a lip balm he reacted to, possibly leading to a ‘booster’ effect.

3) The castor oil reaction might be irritant in nature and not allergic. A dose response in petrolatum might help to resolve this or to at least test castor oil at the percentage found in the cosmetics in question. 

Although almost any allergen, castor oil included, may indeed possess irritant properties as well, the fact that the majority of ~400 patients, and at least 20 control patients, patch tested negative, argues relatively against an irritant reaction.

Furthermore, the reaction pattern, crescendo from D2 to D4, and accompanied by an obvious spreading reaction, is suggestive of an allergic response, and less of an irritant response (although we acknowledge that exceptions exist to this rule).

Finally, resolution of the dermatitis, upon avoidance of castor oil-containing cosmetics, also argues for its relevance as an contact allergen in our patients.

We do agree, that, in order to further clarify the allergenic response, a dose-response patch test may indeed be very useful. Unfortunately, this is considered not feasible as the patients, who’s dermatitis now finally healed, decline further patch testing.

Round 2

Reviewer 1 Report

The manuscript has been substantially improved and is could be accepted in its present form.

However, the description of the ROAT performed by patient 2 could be clearified. Was the ROAT performed for 14 days as stated in the text or was it aborted after 3 days when a positive reaction had appeared?

Author Response

Dear Editor, dear Reviewer,

Thank you for your comments.

To answer your question:

Q: The description of the ROAT performed by patient 2 could be clearified. Was the ROAT performed for 14 days as stated in the text or was it aborted after 3 days when a positive reaction had appeared?

A: The ROAT was aborted when it became positive, that is, after 3 days, although the patient had previously been asked by us to perform it at least 14 days. I agree this is confusing in the text; to avoid confusion, we would rephrase to: "a ROAT gave a positive result after 3 days".

Yours sincerely,

Olivier Aerts